# Knowledge of telemedicine and its associated factors among health professional in Ethiopia: A systematic review and meta-analysis

Adamu Ambachew Shibabaw[1]*, Alex Ayenew Chereka[1], Agmasie Damtew Walle[1], Addisalem Workie Demsash[1], Geleta Nenko Dube[1], Abiy Tasew dubale[1], Sisay Yitayh Kassie[1], Gemeda Wakgari Kitil[2], Mesafint zewold Jember[3], Chernet Desalegn Gebeyehu[4], Aster Temesgen Ariger[3], Eshetie Andargie Dires[5]

1 Department of Health Informatics, College of Health Science, Mattu University, Mettu, Ethiopia, 2 Department of Midwifery, College of Health Science, Mattu University, Mettu, Ethiopia, 3 Department of Health Informatics, Institute of Public Health, College of Medicine and Health Sciences, Dilla University, Dilla, Ethiopia, 4 Department of Biomedical Sciences, College of Health Sciences, Mattu University, Mettu, Ethiopia, 5 Department of Health Informatics, Institute of Health, Bule Hora, Bule Hora, Ethiopia

* adamambachew2@gmail.com

**Data Availability Statement:** All data generated or analyzed is in the manuscript and uploaded as supplementary information.

## Abstract

### Introduction

Telemedicine is a useful tool for decreasing hospital stress, patient suffering, ambulance needs, hospital anxiety, and costs while improving the standard of care. Nonetheless, the lack of awareness regarding telemedicine poses a barrier to its application, presenting several difficulties in underdeveloped nations like Ethiopia. This review evaluates Ethiopian-specific telemedicine knowledge and associated factors.

### Methods

This systematic review was conducted using a search of several online databases in addition to the main databases, like Medline, PubMed, Scopus, and Science Direct. The writers have looked for, reviewed, and summarized information about telemedicine knowledge in the healthcare system. This study contained seven studies that examined telemedicine knowledge in the Ethiopian healthcare sector. Studies that followed the Preferred Reporting Items for Systematic Review and Meta-Analysis Protocols (PRISMA) were found using search engines. The investigation was carried out using STATA version 11. The indicator of heterogeneity ($I^2$) was used to assess the level of heterogeneity among the included studies. The funnel plot was visually inspected, and Egger's regression test was run to check for publication bias. The pooled effect size of every study is estimated using a random-effect model meta-analysis.

### Results

Examination of 2160 studies, seven studies involving 2775 health professionals, and seven out of the 2160 publications assessed satisfied the inclusion criteria and were added to the systematic review and meta-analysis. The pooled prevalence of Telemedicine knowledge

**Funding:** The author(s) received no specific funding for this work.

**Competing interests:** The authors have declared that no competing interests exist.

**Abbreviations:** AA, Addis Ababa; AOR, Adjusted Odd Ratio; CI, Confidence Interval; FBCS, Facility Based Cross-Sectional; FMOH, Ethiopia's Federal Ministry of Health; HP, Health professional; IBCS, Institutional Based Cross-Sectional; TM, Telemedicine; PRISMA, Preferred Reporting Items for Systematic Reviews and Meta-Analyses.

was 45.20 (95% CI: 34.87–55.53). Whereas the pooled factor was computer training was 2.24 times (AOR = 2.24 (95%; CI: 1.64–3.08)), computer access was 2.07 times (AOR = 2.07 (95% CI: 1.50–2.87)), internet access was 3.09 times (AOR = 3.09 (95% CI: 1.34–7.13)), social media access were 3.09 times (AOR = 3.09(95%; CI: 1.34–7.13)), educational status degree and above were 2.73 times (AOR = 2.73; 95% CI: 0.85–8.82), Awareness were 3.18 times (AOR = 3.18 (95%; CI: 1.02–9.91)), Management support was 1.85 (AOR = 1.85 (95% CI: 01.25–2.75)), computer literacy were 2.90 times (AOR = 2.90 (95% CI: 1.81–4.64)), computer owner were 1.70 times (AOR = 1.70 (95% CI: 1.05–2.76)), male gender were 1.95 times (AOR = 1.95 (95% CI: 1.32–2.87)).

## Conclusion

The overall pooled prevalence of telemedicine knowledge was low. Gender, education, management support, computer access, social media access, internet access, telemedicine awareness, and telemedicine training associated with telemedicine knowledge.

## Introduction

Several telemedicine initiatives in sub-Saharan Africa have been established, including the Fundamental of Modern Telemedicine for Africa (FOMTA), the Pan-African e-Network Project, and the Réseauen Afrique Francophone pour la Telemedicine (RAFT) [1]. FOMTA's primary goal is to foster the creation of regional networks connecting research and development centers in developing countries, along with their respective universities, linking them to European countries. Utilizing broadband technology, specifically Integrated Service Digital Network (ISDN), the project aims to facilitate collaboration between developing nations for the generation of new knowledge and the development of innovative and suitable technologies. The ultimate aim of FOMTA is to address unmet healthcare needs and contribute to sustainable economic development in the region [1, 2]

In summary, technological advancements have propelled medical science into a new era, fostering more precise, accessible, and efficient healthcare services while driving innovation in research, diagnostics, and treatment modalities [3].

Telehealth serves to bridge gaps in healthcare access, particularly in underserved or remote areas. It enhances convenience for patients, reduces healthcare costs, and improves the overall efficiency and effectiveness of healthcare delivery [1, 2, 4]. This comprehensive term encompasses various modalities, including telemedicine, teleconsultation, mHealth, and tele monitoring [5, 6]. Telemedicine refers to the utilization of electronic information and communication technologies to deliver clinical services when patients are located at distant locations [7, 8]. "mHealth is a term derived from telehealth, more specifically from telemedicine [9], and is defined as the 'medical and public health practice supported by mobile devices, such as mobile phones, patient-monitoring devices, personal digital assistants (PDAs), and other wireless devices [10]." Telemedicine relies on mHealth applications for supporting effective interaction between healthcare providers and patients [11].

Ethiopia's Federal Ministry of Health (FMOH) is creating a national e-health plan that will include cutting-edge technologies that shift the health industry, advance scientific knowledge of health-related issues, and improve communication between patients and healthcare providers [12]. Innovative technologies like telemedicine affect both the advancement of medical

care and the way of delivering healthcare services [13]. Telemedicine is the use of electronic and information communication technologies to provide a virtual environment that enables remote interaction between healthcare professionals and their patients, and/or among healthcare professionals themselves [6, 14]. Telemedicine plays a significant role in patient-centered healthcare delivery in the diagnosis, and management of chronic diseases and future treatment plane [2, 15].

Telemedicine can make a substantial contribution to the health sector by enabling mainly three applications such like real-time communication between healthcare professionals and patients via video conferences, Store and forward meaning sharing video, voice, image, data, and other medical information with physicians across a distance, and remote patient monitoring which enables medical professionals to monitor a patient remotely using various technological devices [12]. According to a study conducted in hospitals of the north Gondar administrative zone, North West Ethiopia, the proportion of health professional's knowledge of telemedicine was limited [16]. On the contrary, a study conducted on referral hospitals of the Amhara region, North West Ethiopia indicates that the proportion of health professional's knowledge of telemedicine is sufficient but not enough [14, 17]. Studies found that having IT support staff, Socio-demographic factors (gender, residence), Internet availability in the workplace, information sharing culture, computer training, and computer literacy skills are interlinked to knowledge of telemedicine [14, 18–20].

While numerous publications explore the understanding of telemedicine among individuals in both private and public health facilities, our extensive search reveals a lack of a comprehensive systematic review on knowledge about telemedicine and its associated factors in Ethiopia. Therefore, the objective of this systematic review is to ascertain the aggregated level of individual knowledge regarding telemedicine and to identify pertinent influencing factors. The findings of this systematic review aim to provide insights into the overall proportion of individuals' knowledge of telemedicine in Ethiopia. Furthermore, our findings can be valuable for health managers, policymakers, and planners, aiding them in gaining a comprehensive understanding of the collective knowledge of individuals regarding telemedicine in Ethiopia. Additionally, this information may aid them in implementing electronic telemedicine-related information sources and overcoming challenges associated with the adoption of new technology. The outcomes of this study will also support patients, students, and healthcare providers in developing targeted interventions to enhance their understanding of telemedicine.

## Methods

### Source of information and search strategy

Preferred Reporting Items for Systematic Reviews and Meta-Analysis (PRISMA) checklist was used to conduct the research systematically to evaluate Ethiopian healthcare providers' pooled knowledge about telemedicine systems. The research group created a plan for review. From November 12 to December 24, 2023, an online database search was performed. We searched the databases Google Scholar, Medline, PubMed, Embrace, Web of Science, Scopus, ProQuest, CINAHL, Ovid, EBSCO host, and Cochrane Library to find relevant published works. The versions that were used have release dates between November 12, 2023, and December 24, 2023. The papers in online databases were searched systematically using the following combinations of search terms: ("Knowledge") AND ("telemedicine" OR "telemedicine systems" OR "TM") AND ("health professional") AND ("Ethiopia"). The initial search terms that were used were "Knowledge of telemedicine." Once all of the keywords were present in the federated search box, the remaining keywords were added to the major ones one at a time, separated by commas.

## Eligibility criteria

Original research articles from Ethiopia that examined variables associated with Telemedicine Knowledge are included in the analyses. Studies that were freely accessible in full-text English-language publications from peer-reviewed journals or magazines made up this inquiry. Nevertheless, research lacking complete texts, challenging to extract data, non-English publications, uncategorized outcome variables, and studies failing to demonstrate telemedicine knowledge in Ethiopia were eliminated. Furthermore, the study did not include any papers that contained editorial reports, letters, reviews, or commentary.

## Measurements of outcome variables

The results of this systematic review and meta-analysis included an assessment of Ethiopia's collective telemedicine knowledge.

## Data extraction

Two researchers produced a common computer-based spreadsheet using the data from the included studies. A second researcher then verified the spreadsheet for consistency.

## Evaluation of the selected literature's quality

The quality of each study was assessed using a standardized tool that classifies bias potential and can aid in explaining discrepancies in the findings of included studies. Both authors assessed the methodological and other features of each publication using a modified version of the Newcastle Ottawa Scale (NOS) for cross-sectional research, a valid instrument for determining bias risk in observational studies [21]. We determined that articles with nine question, and a Total score (9%) on the modified NOS components were significant after examining a wide range of publications. Furthermore, three authors separately completed a quality control check (S1 Table).

Key: Y = Yes; NR = Not reported, NA = Not appropriate
Question codes.

1. Was the sample frame appropriate to address the target population?

2. Were study participants sampled in an appropriate way?

3. Was the sample size adequate?

4. Were the study subjects and the setting described in detail?

5. Was the data analysis conducted with sufficient coverage of the identified sample?

6. Were valid methods used for the identification of the condition?

7. Was the condition measured in a standard, reliable way for all participants?

8. Was there appropriate statistical analysis?

9. was the response rate adequate, and if not, was the low response rate managed appropriately?

## Data processing and analysis

The recovered data were initially exported from Microsoft Excel and loaded into STATA version 11 to do additional analysis.

# Result

## Search result

There were 2160 articles in all that were found. The title and abstract of 872 titles were checked after eradicating 1009 duplicates, and 279 records were eliminated. Following the full-text screening using the inclusion and exclusion criteria, 272 articles were removed. Among those papers they consider meta-analysis. Finally, 7 papers were included in the study based on the pre-established criteria and quality evaluation (S1 Fig).

## Characteristics of the included studies

This review and meta-analysis included three publications with 2775 respondents. Five of the investigations were conducted in the Amhara area, one in Addis Ababa, and the remaining one in Oromia. Every study that was included was carried out in Ethiopia and employed a cross-sectional study design based on a facility. The study under review discovered that between 31.5% and 65.8% of respondents knew anything about telemedicine. Every study done on health professionals was also included. The Joanna Briggs Institute quality score assessment indicates that all articles meet the specified quality, which was eight and above (S2 Table).

## The pooled magnitude of knowledge of telemedicine in Ethiopia

The four investigations concluded that there was little knowledge of telemedicine in Ethiopia. Health professionals in Ethiopia had a pooled prevalence of 45.20 (95% CI: 34.87–55.53) knowledge of telemedicine systems, according to this meta-data analysis. There was statistically significant heterogeneity, according to a random-effects model (I2 = 0.0%; p = 0.521) (S2 Fig).

These results thus showed that subgroup analysis was not necessary and that there was insignificant heterogeneity among the primary studies.

## Publication bias

An examination of the asymmetry in a funnel plot was used to visually analyze the presence or absence of publication bias. A visual examination of the funnel plot demonstrates that all of the studies were contained within the triangle and further suggests the lopsided distribution. Because of this, the funnel plots of the meta-analysis's results showed that there was no publication bias in any of the included papers (S3 Fig).

## Factors associated with knowledge of telemedicine among health professionals

To implement a telemedicine system, this study looked at several factors of Ethiopia's telemedicine understanding. A total of seven research were used to assess the telemedicine relationship knowledge. As a result, health professionals who had received computer training were 2.26 times more likely to be familiar with telemedicine than those who had not taken computer (AOR = 2.24 (95%; CI: 1.64–3.08). Furthermore, three investigations were carried out to investigate the correlation between computer access and knowledge of telemedicine. The results showed that participants with computer access had a 2.07-fold higher likelihood of knowing about telemedicine compared to those without computer access (AOR = 2.07 (95% CI: 1.50–2.87)). In a similar, three investigations were carried out to assess the relationship between internet access and telemedicine knowledge. The results showed that participants with internet

access had 3.09 times (AOR = 3.09 (95% CI: 1.34–7.13)) as compared to participants who did not have a chance of internet access. Indicates that health professionals with strong social media access were 3.09 times (AOR = 3.09(95%; CI: 1.34–7.13)) more likely to be knowledgeable about telemedicine than those with limited social media access. Three studies investigations the relationship between awareness and telemedicine knowledge, so health professionals who had awareness of telemedicine were 3.18 times more likely to be familiar with telemedicine than those who were unaware of telemedicine had 3.18 times (AOR = 3.18 (95%; CI: 1.02–9.91). One study showed that telemedicine knowledge has a correlation with an educational status degree and above 2.73 times (AOR = 2.73; 95% CI: 0.85–8.82) higher health professionals with an educational status below degree and above. One study showed that telemedicine knowledge correlates with management support was 1.85 times (AOR = 1.85; 95% CI: 01.25–2.75) higher than health professionals for those who didn't have management support. One study showed that telemedicine knowledge correlates with computer literacy was 2.90 times (AOR = 2.90; 95% CI: 1.81–4.64) higher than health professionals for those who were computer illiterate. One study showed that telemedicine knowledge correlates with computer owners were 1.70 times (AOR = 1.70; 95% CI: 1.05–2.76) higher than for health professionals who didn't have a computer. Two studies have shown that telemedicine knowledge correlates with the Male gender were 1.95 times (AOR = 1.95; 95% CI: 1.32–2.87) higher than for health workers for those who were female **(S4 Fig)**.

## Discussion

This study used a systematic review and meta-analysis to evaluate the TM knowledge of Ethiopian healthcare providers. According to our study, healthcare workers' overall TM knowledge was 45.20 (95% CI: 34.87–55.53). Compared to research done in Australia (60%) [22], Saudi Arabia (54.9%)(8), Nepal(77.4%) [23], Sri Lanka (76.4%) [24], the current meta-analysis was less extensive. The context of the study, the backgrounds of the participants, or the variations in the infrastructure of information and communication technology across the countries could all be contributing factors to this variation. Furthermore, Ethiopia is developing emerging technologies at a modest level. However, the system was already in use when those nations were being studied. The outcome of using alternative instruments, a different time frame for the study, or an alternative sample strategy. Furthermore, the everyday exposure of medical practitioners to the global digital environment may have a greater role. This investigation is more extensive than the one conducted in Nigeria [25]. According to this study, healthcare workers with prior computer experience were more likely than those without to have a solid understanding of TM [2, 26]. Bolstered by the notion that computer training will contribute to an increase in TM compared to health professionals without computer access, those with access to one were more likely to be well-versed in telemedicine [26, 27]. This might involve computer access to help medical professionals learn more TM compared to health professionals without access to social media, those with social media usage were more likely to be well-versed in telemedicine [26, 28]. Access to social media promotes knowledge TM compared to health professionals who were uninformed of telemedicine, those who were aware of it were more likely to have a solid knowledge of it [17, 29]. This might be conscious of nearly knowing about TM. Compared to health professionals with inadequate management assistance, those with strong management support were more likely to be well-versed in telemedicine [12]. The fact that management provides support for knowledge transfer Compared to health professionals without personal computers, those with computers were more likely to be well-versed in telemedicine [6, 26, 27]. The fact that some owned computers are being used for internet access may serve as evidence for this. Compared to health professionals with low computer

literacy, those with higher literacy were more likely to be well-versed in telemedicine [6, 15, 27]. This may be corroborated by the observation that computer-literate people will find it easier to learn about telemedicine. Compared to health professionals without telemedicine training, those with telemedicine training were more likely to have solid telemedicine expertise [6, 13, 30]. When compared to health professionals without computer accessibility, those with it were more likely to be well-versed in telemedicine [16, 27]. Compared to health professionals with lower educational levels, those with higher education levels were more likely to have solid knowledge of telemedicine [31, 32]. Compared to female health professionals, male health professionals were more likely to have solid telemedicine expertise [33, 34]. This might have to do with the fact that male health professionals have greater positions inside the company and can thus access new technologies and training.

## Strengths and limitations of the study

A meta-analysis of studies from different regions of Ethiopia provides a summary of the state of telemedicine in the country. Understanding this will be useful for medical and educational institutions, government and health professionals in resource-constrained situations. This study has some drawbacks, the main among which most of the researched articles were cross-sectional study, indeed.

## Conclusion

According to the study, the majority of medical practitioners knew very little about telemedicine. For this reason, the Ethiopian government should prioritize providing management assistance, computer and internet access, and training in telemedicine.

### Recommendation

The Ethiopian government needs to take action to support health professionals in developing their telemedicine training. It should also give health professionals more possibilities to seek higher education since this will increase their awareness of and familiarity with telemedicine. Telemedicine, along with computer training, can drastically change how health professionals understand telemedicine.

## Supporting information

**S1 Checklist. Preferred Reporting Items for Systematic Reviews and Meta-Analyses (PRISMA) checklist.**
(DOCX)

**S1 Fig. Flow chart of study selection for systematic review and meta-analysis knowledge of TM and associated factors among health professionals in Ethiopia, 2023.**
(TIFF)

**S2 Fig. Forest plot of the pooled knowledge of TM among health professionals in Ethiopia, 2023.**
(TIFF)

**S3 Fig. Graphic representation of publication bias using funnel plots of all included studies, 2023.**
(TIFF)

**S4 Fig. Associated factor of studies included in meta-analysis knowledge of TM among health professionals in Ethiopia, 2023.**
(TIFF)

**S1 Table. Quality assessment knowledge of telemedicine and its associated factors among health professional in Ethiopia: A systematic review and meta-analysis.**
(DOCX)

**S2 Table. Descriptive summary of primary studies included in the meta-analysis knowledge of TM and associated factors among health professionals, 2023.**
(DOCX)

## Acknowledgments

The authors express their gratitude to the authors of the original papers that were incorporated and utilized as a source of data for this systematic review and meta-analysis.

## Author Contributions

**Conceptualization:** Adamu Ambachew Shibabaw, Alex Ayenew Chereka, Agmasie Damtew Walle, Addisalem Workie Demsash, Geleta Nenko Dube, Abiy Tasew dubale, Sisay Yitayh Kassie, Gemeda Wakgari Kitil, Mesafint zewold Jember, Chernet Desalegn Gebeyehu, Aster Temesgen Ariger, Eshetie Andargie Dires.

**Data curation:** Adamu Ambachew Shibabaw, Alex Ayenew Chereka, Agmasie Damtew Walle, Addisalem Workie Demsash, Geleta Nenko Dube, Abiy Tasew dubale, Sisay Yitayh Kassie, Gemeda Wakgari Kitil, Mesafint zewold Jember, Chernet Desalegn Gebeyehu, Aster Temesgen Ariger, Eshetie Andargie Dires.

**Formal analysis:** Adamu Ambachew Shibabaw, Agmasie Damtew Walle, Abiy Tasew dubale, Mesafint zewold Jember, Chernet Desalegn Gebeyehu.

**Funding acquisition:** Adamu Ambachew Shibabaw, Aster Temesgen Ariger.

**Investigation:** Adamu Ambachew Shibabaw, Alex Ayenew Chereka, Agmasie Damtew Walle, Addisalem Workie Demsash, Geleta Nenko Dube, Abiy Tasew dubale, Sisay Yitayh Kassie, Gemeda Wakgari Kitil, Mesafint zewold Jember, Chernet Desalegn Gebeyehu, Aster Temesgen Ariger, Eshetie Andargie Dires.

**Methodology:** Adamu Ambachew Shibabaw, Alex Ayenew Chereka, Agmasie Damtew Walle, Addisalem Workie Demsash, Geleta Nenko Dube, Abiy Tasew dubale, Sisay Yitayh Kassie, Gemeda Wakgari Kitil, Mesafint zewold Jember, Chernet Desalegn Gebeyehu, Aster Temesgen Ariger, Eshetie Andargie Dires.

**Project administration:** Adamu Ambachew Shibabaw.

**Resources:** Adamu Ambachew Shibabaw, Alex Ayenew Chereka, Geleta Nenko Dube.

**Software:** Adamu Ambachew Shibabaw, Agmasie Damtew Walle, Addisalem Workie Demsash, Geleta Nenko Dube, Abiy Tasew dubale, Sisay Yitayh Kassie.

**Supervision:** Adamu Ambachew Shibabaw, Geleta Nenko Dube.

**Validation:** Adamu Ambachew Shibabaw, Alex Ayenew Chereka, Agmasie Damtew Walle, Addisalem Workie Demsash, Geleta Nenko Dube, Abiy Tasew dubale, Sisay Yitayh Kassie, Gemeda Wakgari Kitil, Mesafint zewold Jember, Chernet Desalegn Gebeyehu, Aster Temesgen Ariger, Eshetie Andargie Dires.

**Visualization:** Adamu Ambachew Shibabaw, Addisalem Workie Demsash, Abiy Tasew dubale, Sisay Yitayh Kassie, Gemeda Wakgari Kitil, Mesafint zewold Jember, Chernet Desalegn Gebeyehu, Aster Temesgen Ariger, Eshetie Andargie Dires.

**Writing – original draft:** Adamu Ambachew Shibabaw, Addisalem Workie Demsash, Abiy Tasew dubale, Sisay Yitayh Kassie, Gemeda Wakgari Kitil, Chernet Desalegn Gebeyehu, Aster Temesgen Ariger, Eshetie Andargie Dires.

**Writing – review & editing:** Adamu Ambachew Shibabaw, Alex Ayenew Chereka, Abiy Tasew dubale, Sisay Yitayh Kassie, Gemeda Wakgari Kitil, Mesafint zewold Jember, Chernet Desalegn Gebeyehu, Aster Temesgen Ariger, Eshetie Andargie Dires.

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
