## [Decision Letter · Decision Letter 0]

6 Feb 2024

PONE-D-23-43594Knowledge of telemedicine and its associated factors among health professional in Ethiopia: systematic review and Mata-analysis.PLOS ONE

Dear Dr. Shibabaw,

Thank you for submitting your manuscript to PLOS ONE. After careful consideration, we feel that it has merit but does not fully meet PLOS ONE’s publication criteria as it currently stands. Therefore, we invite you to submit a revised version of the manuscript that addresses the points raised during the review process.

We look forward to receiving your revised manuscript.

Kind regards,

Jahanpour Alipour, Ph.D.

Academic Editor

PLOS ONE

2. We note that your Data Availability Statement is currently as follows: [there is no copy of data in this manscriopt it original work]

Reviewers' comments:

Reviewer's Responses to Questions

**Comments to the Author**

1. Is the manuscript technically sound, and do the data support the conclusions?

Reviewer #1: Yes

Reviewer #2: Yes

2. Has the statistical analysis been performed appropriately and rigorously? 

Reviewer #1: I Don't Know

Reviewer #2: Yes

3. Have the authors made all data underlying the findings in their manuscript fully available?

Reviewer #1: Yes

Reviewer #2: Yes

4. Is the manuscript presented in an intelligible fashion and written in standard English?

Reviewer #1: Yes

Reviewer #2: Yes

5. Review Comments to the Author

Reviewer #1: The manuscript examines a significant issue: one of the key barriers to the use of telehealth is doctors' lack of knowledge. While the structure and writing are appropriate, further investigation is required for some cases, as explained below.

• Two different abstracts are presented, which differ in their results section.

• It is recommended to add Telehealth to the keywords.

• The introduction requires significant editing.

• It is recommended to focus on the widespread use of telemedicine in various fields, especially in countries with limited resources, in the second paragraph. If desired, use the following reference. Hayavi-Haghighi MH, Alipour J. Applications, opportunities, and challenges in using Telehealth for burn injury management: A systematic review. Burns. 2023 Sep;49(6):1237-1248. doi: 10.1016/j.burns.2023.07.001. Epub 2023 Jul 13. PMID: 37537108.

• The second paragraph of the introduction presents the findings of the current study, therefore it is more appropriate for the discussion section. The last paragraph of the introduction, especially after stating the study's objective, contains information that is not suitable for this section and is somewhat appropriate for the conclusion or recommendations section.

• The process of scoring based on NOS and the number of questions will be explained.

• It is worth noting that the accuracy of the results may be impacted by the limited number of selected papers. It would be beneficial for the author to provide an explanation for this and address any corresponding shortcomings.

• It appears that there may have been some issues with the search process. It is worth noting that if Ethiopia is a key search term, as stated in the method, studies conducted outside of the country should not have been included in the initial search.

Reviewer #2: The authors have done a great job. A meta-analysis of studies from different regions of Ethiopia provides a summary of the state of telemedicine in the country. Understanding this will be useful for medical and educational institutions, government and health professionals.

However, there are some comments that need to be corrected:

line 95 "TM" is repeated. There is no need to include "TM" twice into the query.

line 177 and 178 - word "Awareness" should it started from lowcase letter ("awareness")?

line 196-197 please give some examples of % numbers in other contries.

The quality of the Figures 1-4 should be improved!

Figure 1. and Figure 3. - Notes under the figures are unreadable

Figure 2. and Figure 4. - difficult or impossible to read words and digits.

6. PLOS authors have the option to publish the peer review history of their article (what does this mean?). If published, this will include your full peer review and any attached files.

Reviewer #1: **Yes: **Mohamma Hosein Hayavi-Haghighi

Reviewer #2: **Yes: **Lukianova Elena

---

## [Author Response · Author response to Decision Letter 0]

22 Feb 2024

Date: February 13, 2023

Dear editorial board member (s) of PLOS ONE journal

We have been recalled to revise the manuscript entitled “Knowledge of telemedicine and its associated factors among health professional in Ethiopia: a systematic review and Meta-analysis”, and identified by unique submission and revised Manuscript Number: PONE-D-23-43594  under PLOS ONE journal, which was submitted for publication. So, we received the editor(s)’ comments for the betterment of the manuscript before its publication.

Thank you editor for your comments, suggestion, directions, and questions. We are very happy in receiving constructive and invaluable comments for the betterment of the manuscript. Accordingly, we have considered all the comments, questions, directions, and suggestions and provided a point-by-point response letter. 

Finally, we have submitted all the required documents in their revised form. We hope that we have addressed all the suggestions, directions, and raised questions and if you believe that point(s) is not addressed, please let us know. 

Thank you very much all editor (s)

On the behalf of the authors

Yours sincerely,

Correspondence author

Point-by-point response letter 

Editor comments

Editor the following changes must be made to your manuscript before moving on to the next stage.

1. Please ensure that your manuscript meets PLOS ONE's style requirements, including those for file naming. The PLOS ONE style templates can be found at.

Author’s Response: Thank you so much. Now it is corrected as plos one’s guideline .

2. We note that your Data Availability Statement is currently as follows: [there is no copy of data in this manuscript it original work

Author’s Response: Thank you so much. We have attached the data.

Author’s Response: Thank you so much. We have update the abstract.

Review Comments to the Author

Author’s Response: Thank you so much. No concern of dual publication.

Reviewer #1: The manuscript examines a significant issue: one of the key barriers to the use of telehealth is doctors' lack of knowledge. While the structure and writing are appropriate, further investigation is required for some cases, as explained below.

• Two different abstracts are presented, which differ in their results section.

Author’s Response: Thank you so much. We have update the abstract.

• It is recommended to add Telehealth to the keywords.

Author’s Response: Thank you so much. We have added Telehealth to the keywords.

•The introduction requires significant editing.

Author’s Response: Thank you so much. We have modified the introduction. 

• It is recommended to focus on the widespread use of telemedicine in various fields, especially in countries with limited resources, in the second paragraph. If desired, use the following reference. Hayavi-Haghighi MH, Alipour J. Applications, opportunities, and challenges in using Telehealth for burn injury management: A systematic review. Burns. 2023 Sep; 49(6):1237-1248. doi: 10.1016/j.burns.2023.07.001. Epub 2023 Jul 13. PMID: 37537108.

Author’s Response: Thank you so much. We have modified the introduction based on your recommendation article. 

• The process of scoring based on NOS and the number of questions will be explained.

• It is worth noting that the accuracy of the results may be impacted by the limited number of selected papers. It would be beneficial for the author to provide an explanation for this and address any corresponding shortcomings.

Author’s Response: Thank you so much. We have NOS number of questions the score has be explained in the document.

• It appears that there may have been some issues with the search process. It is worth noting that if Ethiopia is a key search term, as stated in the method, studies conducted outside of the country should not have been included in the initial search.

Author’s Response: Thank you so much. We have remove Ethiopia in the keyword.

Reviewer #2: The authors have done a great job. A meta-analysis of studies from different regions of Ethiopia provides a summary of the state of telemedicine in the country. Understanding this will be useful for medical and educational institutions, government and health professionals.

Author’s Response: Thank you so much, for your suggestion

However, there are some comments that need to be corrected: line 95 "TM" is repeated. There is no need to include "TM" twice into the query. Line 177 and 178 - word "Awareness" should it started from low case letter ("awareness")? Line 196-197 please give some examples of % numbers in other countries. The quality of the Figures 1-4 should be improved! Figure 1. And Figure 3. Notes under the figures are unreadable Figure 2. And Figure 4. - Difficult or impossible to read words and digits.

Author’s Response: 

1. Line 95 "TM" is repeated.

Author’s Response: Thank you so much. Now TM has removed.

2. There is no need to include "TM" twice into the query.

Author’s Response: Thank you so much. Now is used once.

3. Line 177 and 178 - word "Awareness" should it started from low case letter ("awareness")?

Author’s Response: Thank you so much. Now corrected as you asked.

4. Line 196-197 please give some examples of % numbers in other countries.

Author’s Response: Thank you so much. Now corrected as you asked, and the % of each country is written.

5. The quality of the Figures 1-4 should be improved! Figure 1. And Figure 3. Notes under the figures are unreadable Figure 2. And Figure 4. - Difficult or impossible to read words and digits.

6. Author’s Response: Thank you so much. Now all the figure 1-figure 4 are improved we have analyzed again and the figure is improved.

With regards!

Thank you!

Yours sincerely 

Correspondence author

---

## [Decision Letter · Decision Letter 1]

11 Mar 2024

Knowledge of telemedicine and its associated factors among health professional in Ethiopia: systematic review and Mata-analysis.

PONE-D-23-43594R1

Dear Adamu Ambachew Shibabaw,

We’re pleased to inform you that your manuscript has been judged scientifically suitable for publication and will be formally accepted for publication once it meets all outstanding technical requirements.

Kind regards,

Jahanpour Alipour, Ph.D.

Academic Editor

PLOS ONE

Additional Editor Comments (optional):

Reviewers' comments:

Reviewer's Responses to Questions

**Comments to the Author**

1. If the authors have adequately addressed your comments raised in a previous round of review and you feel that this manuscript is now acceptable for publication, you may indicate that here to bypass the “Comments to the Author” section, enter your conflict of interest statement in the “Confidential to Editor” section, and submit your "Accept" recommendation.

Reviewer #1: All comments have been addressed

Reviewer #2: All comments have been addressed

2. Is the manuscript technically sound, and do the data support the conclusions?

Reviewer #1: Yes

Reviewer #2: Yes

3. Has the statistical analysis been performed appropriately and rigorously? 

Reviewer #1: Yes

Reviewer #2: Yes

4. Have the authors made all data underlying the findings in their manuscript fully available?

Reviewer #1: Yes

Reviewer #2: Yes

5. Is the manuscript presented in an intelligible fashion and written in standard English?

Reviewer #1: Yes

Reviewer #2: Yes

6. Review Comments to the Author

Reviewer #1: The authors have done a very good job of editing the manuscript and correcting all the errors. Of course, many sources still need to be checked. Good luck to the authors

Reviewer #2: (No Response)

7. PLOS authors have the option to publish the peer review history of their article (what does this mean?). If published, this will include your full peer review and any attached files.

Reviewer #1: **Yes: **Mohammad Hosein Hayavi-Haghighi

Reviewer #2: **Yes: **Lukianova Elena

---

## [Editor Report · Acceptance letter]

25 Mar 2024

PONE-D-23-43594R1 

PLOS ONE

Dear Dr. Shibabaw, 

I'm pleased to inform you that your manuscript has been deemed suitable for publication in PLOS ONE. Congratulations! Your manuscript is now being handed over to our production team.

Kind regards, 

on behalf of

Dr., Jahanpour Alipour 

Academic Editor

PLOS ONE